# PALMER: Perception-Action Loop with Memory for Long-Horizon Planning

**Onur Beker** **Mohammad Mohammadi** **Amir Zamir**

Swiss Federal Institute of Technology (EPFL)

## Abstract

To achieve autonomy in a priori unknown real-world scenarios, agents should be able to: i) act from high-dimensional sensory observations (e.g., images), ii) learn from past experience to adapt and improve, and iii) be capable of long horizon planning. Classical planning algorithms (e.g. PRM, RRT) are proficient at handling long-horizon planning. Deep learning based methods in turn can provide the necessary representations to address the others, by modeling statistical contingencies between observations. In this direction, we introduce a general-purpose planning algorithm called PALMER that combines classical sampling-based planning algorithms with learning-based perceptual representations. For training these perceptual representations, we combine Q-learning with contrastive representation learning to create a latent space where the distance between the embeddings of two states captures how easily an optimal policy can traverse between them. For planning with these perceptual representations, we re-purpose classical sampling-based planning algorithms to retrieve previously observed trajectory segments from a replay buffer and restitch them into approximately optimal paths that connect any given pair of start and goal states. This creates a tight feedback loop between representation learning, memory, reinforcement learning, and sampling-based planning. The end result is an experiential framework for long-horizon planning that is significantly more robust and sample efficient compared to existing methods.

## 1 Introduction

Animals and humans operate on high-dimensional stimuli (e.g., vision) to achieve diverse and ever-changing goals necessary for their survival [1, 2, 3, 4, 5]. Learning through trial-and-error plays a fundamental role in this [6, 7, 8, 9, 10, 5]. Even in simplest environments, a brute-force approach to trial-and-error by trying every possible action for achieving every possible goal is intractable. The complexity of this search motivates memory-based mechanisms for compositional thinking. Examples of such mechanisms include : i) remembering relevant segments of past experience, ii) recomposing them in new counterfactual ways to form plans, and iii) executing such plans as part of a targeted search strategy. Such mechanisms for recycling past successful behavior can significantly accelerate trial-and-error compared to uniformly sampling all possible actions. This is because the same behavior (i.e., sequence of actions) can remain valid for different goals and in different contexts, due to the inherent compositional structure of real-world goals as well as the commonality of the physical laws that govern real-world environments.

What principles can allow for memory mechanisms to remember and recompose bits of experience? The concept of dynamic programming (DP) is directly related to this discussion, as it greatly reduces the computational cost of trial-and-error by employing the principle of optimality [11]. This principle can be colloquially stated as treating new and complex problems as a recomposition of old and simpler sub-problems that were already solved before. Recent work [12, 13, 14] employs this perspective to

36th Conference on Neural Information Processing Systems (NeurIPS 2022).

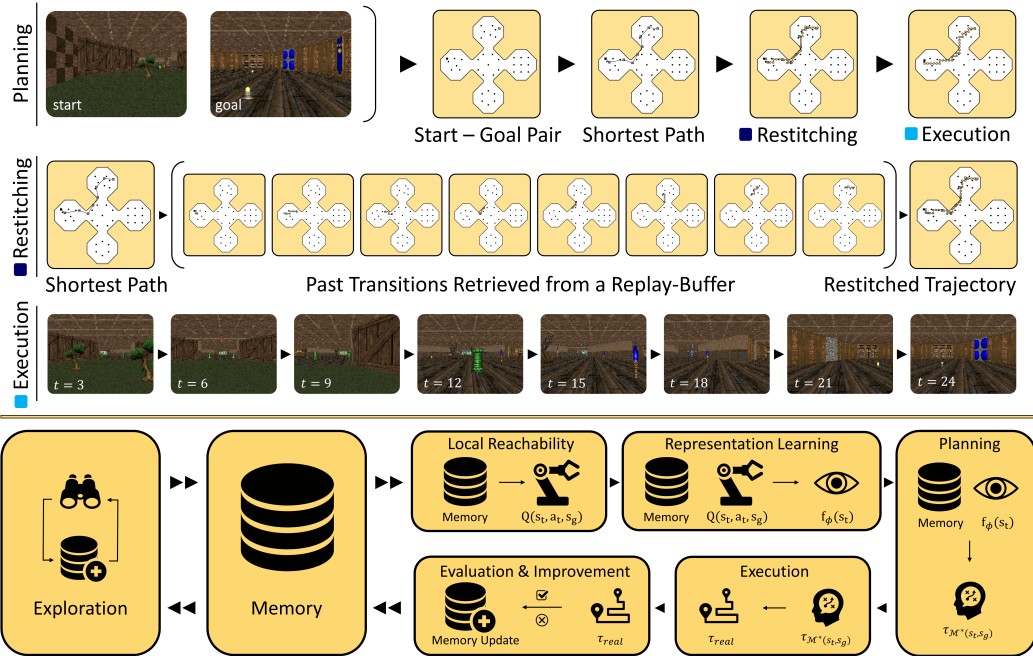

Figure 1: **Top**: Given a start-goal image pair, PALMER plans a path between them by concatenating the endpoints of past trajectory segments retrieved from a provided replay buffer. This is enabled by a state embedding function $f_\phi$ that can identify close-by states, and results in robust long-horizon planning. **Bottom**: To achieve this : i) it uses offline Q-learning to obtain *local reachability* estimates between states, ii) uses these Q-values for *representation learning* to train $f_\phi$, iii) uses $f_\phi$ to *plan* over the replay buffer, iv) *executes* these plans, v) *evaluates* the resulting trajectories and inserts them back into the replay buffer to *improve* its contents.

build hierarchical reinforcement learning (RL) algorithms for goal-reaching tasks. Such methods set edges between states using a distance regression model to build a planning graph, perform shortest path computations over it using DP-based graph search, and follow the resulting shortest paths with a learning-based local policy. Our paper builds upon this line of work.

***Contribution:*** We describe a long-horizon planning method that directly operates on high dimensional sensory input observable by an agent on its own (e.g., images from an onboard camera). Our method combines classical sampling-based planning algorithms with learning-based perceptual representations, to retrieve and recompose previously observed sequences of state transitions in a replay buffer. This is enabled by a two-step process. *First*, we learn a latent space where the distance between two states captures how many timesteps it takes for an optimal policy to go from one to the other. To achieve this, we use goal-conditioned Q-values learned through offline hindsight relabelling [15] for contrastive representation learning. *Second*, we threshold this learned latent distance metric to define a neighborhood criterion between states. We then define sampling-based planning algorithms that search over the replay buffer [12] to retrieve and stitch together trajectory segments (i.e., past sequences of observed transitions) whose endpoints are neighboring states. This trajectory stitching approach allows for creating planning graphs to connect any pair of start and goal states that were observed before (as depicted in Fig.1). Our approach operates on offline unlabeled data, and can therefore be combined with any exploration method to populate the replay buffer. Our experiments implement an image-based navigation policy in simulation, using an offline replay buffer populated with uniform random-walk exploration data.

## 2   Perception-Action Loop with Memory Retrieval[1]

*Nomenclature:* An environment is represented as a tuple $\langle \mathcal{S}, \mathcal{A}, p_{env} \rangle$, where $\mathcal{S}$ and $\mathcal{A}$ are the state and action spaces, and $p_{env}(s'|s, a)$ is the Markovian transition dynamics. A trajectory $\tau \in \mathcal{T}$ is any sequence of states and actions. $\tau_0$ , $\tau_{-1}$ , $\tau_i$ denote the first, last, and $i$'th states in $\tau$ respectively. The length of a trajectory in terms of timesteps is denoted as $len(\tau)$, and concatenation of two

---

[1]Most sub-sections have a corresponding section in the supplementary for further elaboration.

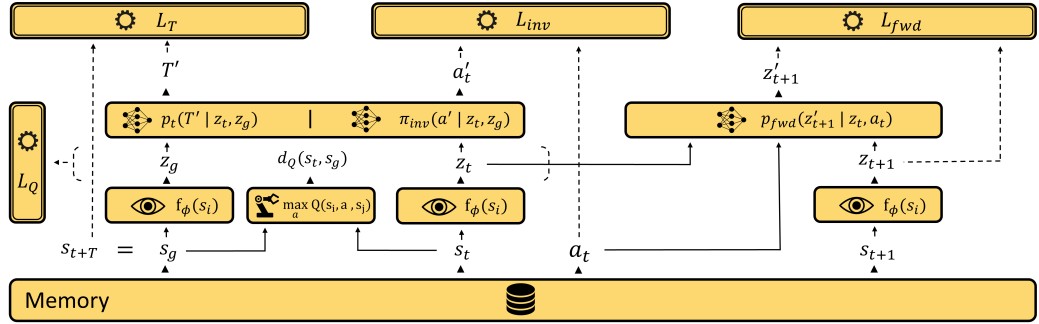

Figure 2: An overview of the functions, inputs, and losses used in our method (see Sec.2.2 for details). We aim to train a perceptual encoder $f_\phi$ with two properties: i) representations of two states should be close if they were observed to be easily reachable from each other within a low number of timesteps, ii) the representation of a state should capture a minimal sufficient statistic to inform an agent about the actions needed to reach nearby states.

trajectories is denoted as $\tau_{cat} = \tau_1 \circ \tau_2$. We assume an additive reward function $\mathcal{R} : \mathcal{T} \to \mathbb{R}$ where $\mathcal{R}(\tau) = \sum_{(s,a)\in\tau} r(s,a)$. We call a finite set of trajectories $\mathcal{M} = \{\tau_i\}$ a replay buffer.

## 2.1 Perceptual Representations that Capture Local Reachability

A key component of our framework is a perceptual encoder $f_\phi(s) : \mathcal{S} \to \mathbb{R}^d$ that maps states into a representation space where L2 distance $d_\phi(s_t, s_g) := \|f_\phi(s_t) - f_\phi(s_g)\|$ captures local reachability (i.e., how many timesteps it takes for the optimal policy to go from one state to another). To discuss this more rigorously, we follow the work of [16, 12] and define a goal-conditioned reward function $r(s_t, a, s_{t+1}, s_g) = -\mathbb{1}_{s_{t+1}\neq s_g}$ that returns $-1$ for all steps before reaching a goal. This means goal-conditioned Q-values [16, 17] for the optimal policy correspond to negative shortest-path distances (i.e., $max_a Q(s_t, a, s_g) = V(s_i, s_j) = -len(\tau_{sp})$). We can then define a symmetric distance metric between states as $d_Q(s_c, s_g) := max(-V(s_c, s_g), -V(s_g, s_c))$. This in turn corresponds to the two-way consistency criterion proposed in [13]. What we want from $f_\phi(s)$ is for $d_\phi(s_c, s_g)$ and $d_Q(s_c, s_g)$ to roughly correlate.

## 2.2 Representation Learning via Reinforcement Learning

Any perceptual encoder $f_\phi$ whose latent representations satisfy the local reachability property defined in Sec.2.1 can be used to implement the nearest neighbor retrieval and trajectory stitching mechanisms for the upcoming sections 2.3 and 2.4. This section discusses *one possible way* to obtain such a perceptual encoder, by using goal-conditioned Q-values for contrastive representation learning.

We propose a model (depicted in Fig.2) that includes the following standard components from the literature: **i)** $z = f_\phi(s)$, projecting a state into a latent representation; **ii)** $p_{fwd}(z'_{t+1} \mid z_t, a_t)$, modelling the transition distribution induced by $p_{env}(s'|s, a)$ over the latent space $z = f_\phi(s)$, as discussed in [18, 19]; **iii)** $\pi_{inv}(a'_t \mid z_t, z_g)$, defining a distribution of actions to reach a goal state, as discussed in [18, 19, 14]; **iv)** $p_t(T' \mid z_t, z_g)$, modelling the distribution of timesteps necessary to reach a goal state, as discussed in [20]; **v)** $Q(s_t, a_t, s_g)$, a Q-value function that provides local reachability estimates between pairs of states, as discussed in [12, 16].

Following [12, 16], we train $Q(s_t, a_t, s_g)$ over an offline replay buffer $\mathcal{M}$, using hindsight relabelling [15, 16] with a reward function $r(s_t, a, s_{t+1}, s_g) = -\mathbb{1}_{s_{t+1}\neq s_g}$. After training $Q(s_t, a_t, s_g)$ in isolation, we freeze its parameters and use it to define a contrastive loss function [21] $L_Q$ as explained below. We then train the remaining components using the same replay buffer $\mathcal{M}$. We randomly sample a transition $(s_t, a_t, s_{t+1})$ and a time difference $T$, and set the goal state as $s_g := s_{t+T}$, as in hindsight relabelling. We then minimize the following losses:

- $L_Q(s_t, s_g) = l_{hinge}(d_\phi(s_t, s_g) - d_p) \, \mathbb{1}_{d_Q(s_t,s_g)\leq c_Q} + l_{hinge}(d_p - d_\phi(s_t, s_g)) \, \mathbb{1}_{d_Q(s_t,s_g)\geq c_Q}$, where $l_{hinge}$ is the hinge loss [22]. This contrastive loss dictates that perceptual representations should be close together (i.e., $d_\phi(s_t, s_g) \leq d_p$ holds) if and only if two states are close to each other in terms of reachability (i.e., $d_Q(s_t, s_g) \leq c_Q$ holds). $d_p$ and $c_Q$ are hyperparameters.

- $L_T(T', T)$, $L_{inv}(a'_t, a_t)$, and $L_{fwd}(z'_{t+1}, z_{t+1})$ are MSE and cross-entropy losses [19, 20]. $L_T$ and $L_{inv}$ dictate that perceptual representations should capture enough information to know when and how an agent can reach from one state to another, while $L_{fwd}$ dictates that they should capture only a minimal-sufficient statistic for doing so ([19] presents a more elaborate discussion).

## 2.3 Perceptual Experience Retrieval (PER)

Given a perceptual encoder $f_\phi$ that captures local reachability, we go over all states $s_i \in \mathcal{M}$ in the replay buffer and compute their projections $z_i = f_\phi(s_i)$, which are stored alongside the states themselves. We then employ $z_i$ to implement two retrieval mechanisms from the replay buffer: i) retrieving neighboring states, and ii) retrieving neighboring trajectories.

*i) Retrieving Neighboring States:* Given a query state $s_c$ and radius $d_p$ (i.e., the same one used in the contrastive loss $L_Q$ in Sec.2.2), retrieving neighboring states amounts to computing the set $\mathcal{N}_{d_p}(s_c) = \{s_n \mid d_\phi(s_c, s_n) \le d_p\}$, which can be achieved by a straightforward L2 distance computation and thresholding. The number of neighbors $|\mathcal{N}_{d_p}(s_c)|$ of a query state $s_c$ is an approximate measure of how many times the agent has visited around $s_c$, which also makes it a good visitation-count that is applicable to both discrete and continuous state spaces.

*ii) Retrieving Neighboring Trajectories:* Given a starting state $s_c$ and a goal state $s_g$, we can search the replay buffer for the highest reward trajectory segment $\tau$ that starts from a state $\tau_0$ in $\mathcal{N}_{d_p}(s_c)$ and ends in a state $\tau_{-1}$ in $\mathcal{N}_{d_p}(s_g)$. This corresponds to the following optimization problem:

$$\tau_{\mathcal{M}(s_c, s_g)} := \arg\max_{\tau \in \mathcal{M}} \mathcal{R}(\tau) \quad \text{s.t.} \quad \tau_0 \in \mathcal{N}_{d_p}(s_c) , \ \tau_{-1} \in \mathcal{N}_{d_p}(s_g) \tag{1}$$

To find $\tau_{\mathcal{M}(s_c, s_g)}$, we first select all state pairs $(s_i, s_j) \in \mathcal{N}_{d_p}(s_c) \times \mathcal{N}_{d_p}(s_g)$. We then take all sequences of transitions $\tau_{ij} = \{s_i, a_i, s_{i+1}, ..., s_{j-1}, a_{j-i}, s_j\}$ that start from $s_i$, end at $s_j$, and are below a length threshold in terms of timesteps. We sort them based on $\mathcal{R}(\tau_{ij})$, and return the trajectory with the highest reward. We call this trajectory retrieval process 'Perceptual Experience Retrieval' (PER). We use PER only to retrieve short trajectory segments between close-by states $(s_c, s_g)$ (i.e., hence the length threshold on $\tau_{ij}$). These are then stitched together into long global trajectories using the planning algorithms defined in the next section.

## 2.4 Long-Horizon Planning Through Stitching Trajectory Segments

This section discusses how PER can be employed for long-horizon planning. Classical sampling-based planning algorithms such as RRT [23] or PRM [24] connect points sampled from obstacle-free space with line segments in order to build a planning graph. We instead reimagine them as memory search mechanisms by altering their subroutines so that whenever an edge is created, a trajectory is retrieved from the replay buffer through PER (eq.1) and stored in that edge. Our new definitions for these subroutines directly mirror the original ones given in [25]:

*1) Sampling:* Sampling originally returns a point from obstacle free space. We instead return a state $s_c$ from the replay buffer $\mathcal{M}$ using any distribution (e.g., uniform, or based on visitation-counts).

*2) Lines and Their Cost:* The equivalent of drawing a line segment in our framework is retrieving a trajectory $\tau_{\mathcal{M}(s_c, s_g)}$, and its length and cost are $len(\tau_{\mathcal{M}(s_c, s_g)})$ and $-\mathcal{R}(\tau_{\mathcal{M}(s_c, s_g)})$ respectively.

*3) Nearest State and Neighborhood Queries:* Given a query point $s_i$, these subroutines return the closest point or a neighborhood of points within a distance, among a set of vertices $V = \{s_j\}$. We preserve these definitions, and only replace the metric from euclidean distance to $len(\tau_{\mathcal{M}(s_c, s_g)})$.

$$Nearest(V, s_g) := \arg\min_{s_c \in V} \ len(\tau_{\mathcal{M}(s_c, s_g)})$$

$$Near(V, s_g, r) := \{s_c \in V \mid len(\tau_{\mathcal{M}(s_c, s_g)}) \le r\}$$

*4) Collision Tests:* Collision tests originally prevent the sampling and line drawing subroutines from intersecting obstacles. Since we are planning in retrospect, any such undesirable event can be handled during PER by adjusting the reward function (i.e., if $\tau$ has such an event, this reflects on $\mathcal{R}(\tau)$).

Using these subroutines directly in-place of their originals, we reimplement experiential equivalents of PRM, RRT, and RRT*, which we call R-PRM, R-RRT, R-RRT*. We denote the resulting planned trajectory as $\tau_{\mathcal{M}^*(s_c, s_g)}$. Algorithms 1, 2 describe R-PRM as an example, and the supplementary contains descriptions for R-RRT, R-RRT*.

---

**Algorithm 1** R-PRM (Roadmap Construction)

---

1: **Input:** $f_\phi, \mathcal{M}$
2: $V \leftarrow \{SampleFree_i\}_{i=1,...,num\_vertices}; \; E \leftarrow \emptyset$ ▷ Initialize vertices and edges
3: **for each** $s_i \in V$ **do**
4:    $U \leftarrow Near(V, s_i, r) \setminus \{s_i\}$
5:    **for each** $s_j \in U$ **do** ▷ Place PER trajectories in edges
6:       $E \leftarrow E \cup \{(s_i, s_j) : \tau_{edge} = \tau_{\mathcal{M}(s_i, s_j)}, \; d_{edge} = -\mathcal{R}(\tau_{\mathcal{M}(s_i, s_j)})\}$
   **return** $G = (V, E)$

---

---

**Algorithm 2** R-PRM (Trajectory Restitching Given the Constructed Roadmap)

---

1: **Input:** $s_c, s_g, G = (V, E), \mathcal{R}(\tau), f_\phi, \mathcal{M}$
2: **for each** $s_i \in V$ **do** ▷ Insert $s_c$ and $s_g$ into the PRM graph
3:    **if** $len(\tau_{\mathcal{M}(s_c, s_i)}) \leq r$ **then** ▷ Place PER trajectories in edges
4:       $E \leftarrow E \cup \{(s_c, s_i) : \tau_{edge} = \tau_{\mathcal{M}(s_c, s_i)}, \; d_{edge} = -\mathcal{R}(\tau_{\mathcal{M}(s_c, s_i)})\}$
5:    **if** $len(\tau_{\mathcal{M}(s_i, s_g)}) \leq r$ **then**
6:       $E \leftarrow E \cup \{(s_i, s_g) : \tau_{edge} = \tau_{\mathcal{M}(s_i, s_g)}, \; d_{edge} = -\mathcal{R}(\tau_{\mathcal{M}(s_i, s_g)})\}$

7: $\tau_{stitched} \leftarrow \emptyset$
8: $\{s_j\} \leftarrow ShortestPath(s_c, s_g, G, \mathcal{R}(\tau))$ ▷ Trajectory stitching by dynamic programming
9: **for** $0 < i < |\{s_j\}|$ **do** ▷ Concatenate PER trajectories along the shortest path
10:    $\tau_{stitched} \leftarrow \tau_{stitched} \circ \tau_{\mathcal{M}(s_{i-1}, s_i)}$
   **return** $\tau_{\mathcal{M}^*(s_c, s_g)} = \tau_{stitched}$

---

We note two things about our proposed planning algorithms. First, they can optimize any general reward function $\mathcal{R}$. As the number of sampled vertices increases, $\mathcal{R}(\tau_{\mathcal{M}^*(s_c, s_g)})$ gets optimized through dynamic programming (i.e., by minimizing the Bellman error between vertices of the roadmap $G$), therefore employing the same mechanism as classical sampling-based planning algorithms [25]. Second, they operate on an offline dataset of unlabeled transitions which solely consists of high-dimensional on-board sensory data (e.g. images), *without assuming any auxiliary instrumentation in the environment or oracle information that cannot be sensed by the agent on its own*. They therefore aim to relax the assumptions classical sampling-based planning methods make about what constitutes a model (e.g., replacing a geometric environment model with sensory experience) and what constitutes a state (e.g., enabling search and planning directly over images).

### 2.5 Refining Memory Contents via Forming and Executing Plans

We iteratively form and execute $\tau_{\mathcal{M}^*(s_c, s_g)}$, and whenever execution is successful, we insert the resulting new trajectories back into $\mathcal{M}$. We note that these new trajectories are not exactly the same as $\tau_{\mathcal{M}^*(s_c, s_g)}$, because $\tau_{\mathcal{M}^*(s_c, s_g)}$ contains approximate mismatches between the endpoints of its stitched trajectory segments due to nearest neighbor retrieval. Forming and executing plans this way creates the following perception-action loop: **i)** $\mathcal{M}$ with refined contents is used to train a more accurate $Q(s_t, a, s_g)$, **ii)** a more accurate $Q(s_t, a, s_g)$ creates a more accurate distance metric $d_\phi$, **iii)** a better $d_\phi$ generates better $\tau_{\mathcal{M}^*(s_c, s_g)}$, **iv)** better $\tau_{\mathcal{M}^*(s_c, s_g)}$ result in higher frequencies of successful execution to further refine $\mathcal{M}$ (see the supplementary for an algorithmic description).

## 3 Related work

*Self-supervised goal reaching:* Our approach is closely related to goal-reaching methods that combine learning-based distance-regression with graph search, particularly Semi-parametric Topological Memory (SPTM) [14] and Search on the Replay Buffer (SoRB) [12], which we compare to in our experiments. The key difference of our approach is that when setting the edges of the planning graph, it retrieves transitions that *actually happened* rather than relying on learned distance regression. This brings two main benefits. First is robustness. Local reachability estimates are susceptible to overestimation when evaluated between pairs of states that are far apart or unreachable. This

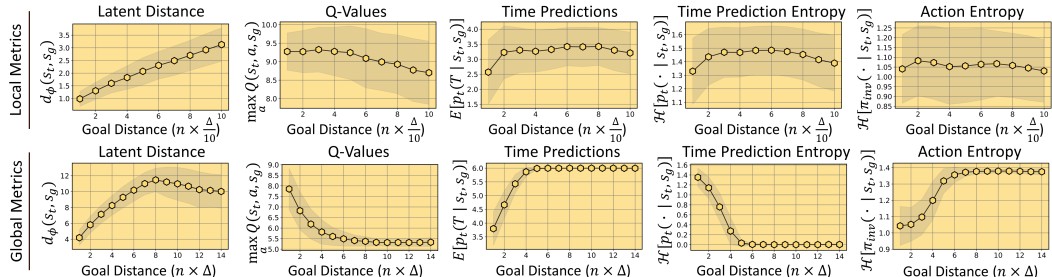

Figure 3: A comparison between perceptual distances $d_\phi$ and other suitable metrics from Sec.2.2. While all of these metrics are reasonably monotonic with physical reachability (i.e., goal distance), only perceptual distances $d_\phi$ do not saturate when evaluated locally (i.e., for close by goals). In addition, the ratio between the variance of $d_\phi$ and the slope of its mean is much smaller compared to other sensible metrics (i.e., $d_\phi$ has a high signal-to-noise ratio). This means that perceptual distances can implement a more accurate nearest-neighbor criterion for perceptual experience retrieval and trajectory stitching, compared to the other metrics.

is because such states rarely occur together and are therefore out of distribution for the distance regression model. This creates 'hallucinated' shortcuts in the planning graph that corrupt shortest path queries [12, 13]. To address this, [14] employs temporally consistent localization and adaptive waypoint selection, while [12] employs distributional Q-learning and an ensemble of Q-functions. In our approach, eq.1 naturally addresses this problem, since it requires an actual short trajectory in the dataset approximately connecting two states before marking them as close. The second benefit of our approach is that *it can optimize general reward functions*. This is because it decouples the reachability metric $len(\tau)$ (used in nearest neighbor queries and as a threshold to create edges) from the downstream task reward $\mathcal{R}(\tau)$ (used to set edge distances), unlike previous work.

*Image-Based Navigation:* [26, 27, 28] present learning-based navigation systems that incrementally build roadmaps through online operation. Our approach has two main differences: **i)** it builds a roadmap entirely using raw offline data, therefore allowing applications like multi-robot learning without additional loop-closure mechanisms to fuse graphs from multiple agents, **ii)** our approach can optimize general reward functions, therefore it is not limited to navigation.

*Robot Motion Planning:* A common approach to motion planning is to first run a sampling-based planning algorithm [29, 25], and then refine the result through trajectory optimization [30, 31, 32] to satisfy constraints [33, 34, 35]. An important bottleneck is that sampling-based planning algorithms require a precomputed map of the environment, and our approach extends such algorithms in a way that relaxes this requirement by replacing a precomputed map with raw exploration experience.

*SLAM and Geometric Maps:* SLAM based methods [36] can autonomously construct high-fidelity geometric maps [37, 38], therefore alleviating the bottleneck of precomputing environment maps. The downside of such approaches is that they can abstract away useful physical and semantic affordances. For example, a purely geometric map cannot plan a path through a traversable field of tall-grass, while our approach can learn such affordances as long as they are represented in past experiences.

# 4   Experiments[2]

*Setup:* Our experiments are performed in ViZDoom [39], Habitat [40], and the Maze2D benchmark [41]. The VizDoom environment consists of a clover shaped maze. States solely consist of four images $I_{North/East/South/West}$ that form a panorama (i.e., $4 \times 3 \times 160 \times 120$ dimensions), and actions move the agent North/South/East/West by a fixed distance $\Delta$. The maze contains many long-thin column-like obstructions (shown as dots in visualizations). Habitat experiments contain demonstrations on two large-scale scans of real-world apartments: i) Roxboro, with a total area of 62 m2, and ii) Annawan, which has a total-area of 75$m2$. States consist of a single 150 FOV image (i.e., $3 \times 256 \times 256$ dimensions). There are 3 actions: $\{turn\_left\_30\_deg, turn\_right\_30\_deg, move\_forward\_\Delta\}$. Maze2D is a continuous control task, where states consist of the 2D position and velocity of a point mass, and actions correspond to 2D accelerations. In all environments, an offline training dataset is collected by a uniform random walk exploring the environment. For VizDoom and Habitat, this offline training dataset consists of only 300k and 150k timesteps respectively, while for Maze2D there are 1e6 timesteps. Supplementary material contains further details.

---

[2]All experiments have a corresponding section in the supplementary providing further implementation details.

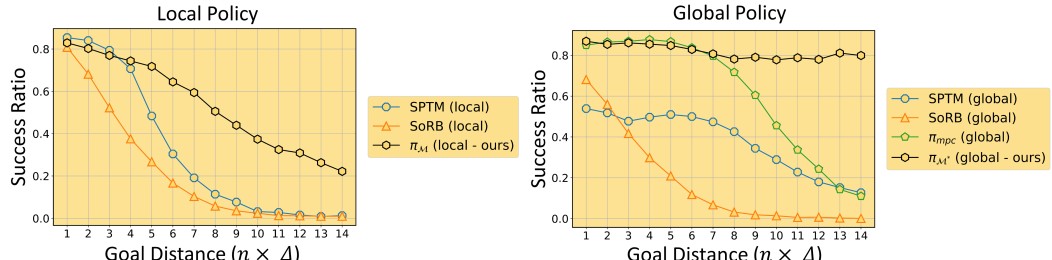

Figure 4: Comparisons of our local policy $\pi_\mathcal{M}$ and global policy $\pi_{\mathcal{M}^*}$ with SPTM and SoRB. $\pi_\mathcal{M}$ performs well because it avoids getting stuck (as such events are filtered by eq.1), while $\pi_{\mathcal{M}^*}$ performs well because it builds robust roadmaps without hallucinated shortcuts; therefore avoiding the main failure modes of the baselines.

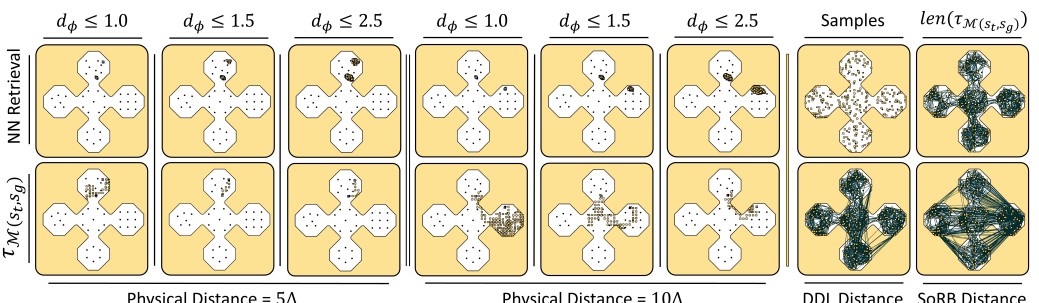

Figure 5: At the core of PALMER is a process called *perceptual experience retrieval* (PER). Given a query pair of current-goal states, PER searches the replay buffer to retrieve the highest scoring trajectory $\tau_{\mathcal{M}(s_t, s_g)}$ whose first and last states are close to the query pair according to the perceptual distance $d_\phi$. **Left, Middle**: Visualizations of $\tau_{\mathcal{M}(s_t, s_g)}$ retrieved using PER and nearest neighbor states $\mathcal{N}_{d_p}(s_t)$ retrieved using $d_\phi$. **Right**: Setting edges of a roadmap using $len(\tau_{\mathcal{M}(s_t, s_g)})$, compared with distance estimates used in SORB and DDL [20]. We found that distance estimates from baselines are prone to setting false edges that cross map boundaries.

## 4.1 Experiments in Vizdoom

*Validating Perceptual Representations:* Fig.3 shows that $d_\phi(s_t, s_g)$ obtained from our model captures a suitable notion of local reachability. Fig.5 in turn shows that retrieving nearest neighbor states $\mathcal{N}_{d_p}(s_t)$ from $\mathcal{M}$ using $d_\phi$ (i.e., NN retrieval) returns physically close states.

*Validating Perceptual Experience Retrieval (PER):* Fig.5 shows visualizations of trajectories retrieved with PER. We implement a retrieval policy $\pi_\mathcal{M}$ that computes $\tau_{\mathcal{M}(s_t, s_g)}$ through eq.1 at each timestep $t$ and executes $argmax_a\ Q(s_t, a, \tau_{\mathcal{M}(s_t, s_g), s, 1})$, therefore forming a model predictive control (MPC) loop. We evaluate $\pi_\mathcal{M}$ in an image-based navigation task where start/goal images are sampled randomly to have an euclidean distance $n \times \Delta$ in between, and a trial is considered successful if the agent can get within $\Delta$ proximity of the goal position within $4 \times n$ time-steps. We use the local policies from SORB [12] and SPTM [14] as baselines. Fig.4 shows the results. The main mode of failure for both SPTM and SORB local policies is that they get stuck in column-like structures. $\pi_\mathcal{M}$ avoids this, since eq.1 retrieves collision free $\tau_{\mathcal{M}(s_t, s_g)}$.

*Robust Distances:* PER also helps avoid hallucinations in local distance regression. Fig.5 illustrates this point by setting edges between sampled states by thresholding $len(\tau_{\mathcal{M}(s_t, s_g)})$, where methods of [12, 20] are used as baselines. It can be seen that edges set by $len(\tau_{\mathcal{M}(s_c, s_g)})$ are more robust.

*Proposed Planning Algorithms:* Fig.6 shows visualizations of planning graphs and $\tau_{\mathcal{M}^*(s_c, s_g)}$ produced by R-PRM, R-RRT, and R-RRT*. It can be seen that R-PRM doesn't contain any hallucinated edges, while R-RRT and R-RRT* maintain the visual characteristics of their classical counterparts (i.e., R-RRT has jagged branches with uniform coverage, while R-RRT* has straight branches shooting out from the root). We implement an MPC policy $\pi_{\mathcal{M}^*}$ that replans at each timestep $t$ using Algorithm 2 to return $\tau_{\mathcal{M}^*(s_t, s_g)}$, and executes $argmax_a\ Q(s_t, a, \tau_{\mathcal{M}^*(s_t, s_g), s, 1})$. We again

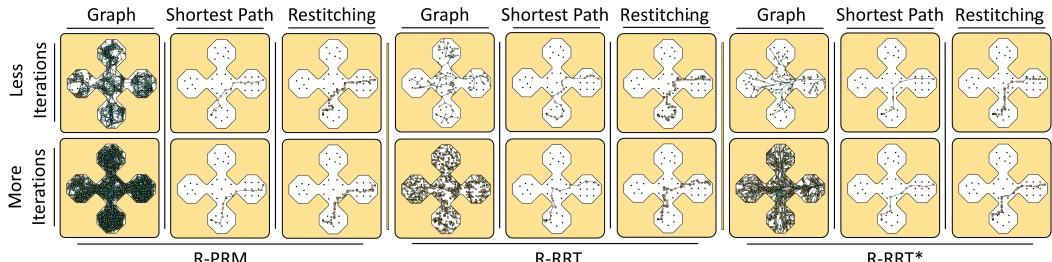

Figure 6: We repurpose conventional sampling-based planning algorithms as memory search mechanisms, by altering their graph building subroutines so that whenever an edge is created a trajectory $\tau_{\mathcal{M}(s_t, s_g)}$ is retrieved through PER and stored in that edge. We visualize the resulting planning graphs produced by our proposed algorithms R-PRM, R-RRT, R-RRT*.

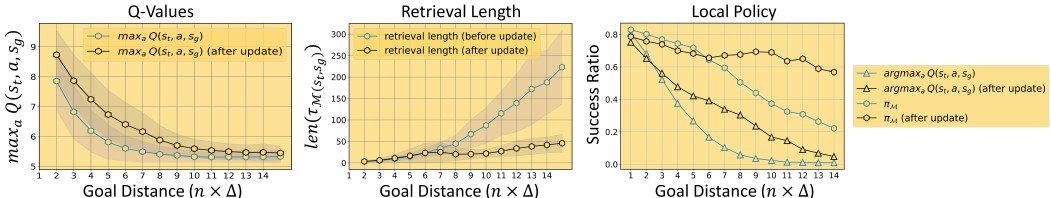

Figure 7: Memory Refinement: In PALMER, a policy has three groups of parameters: $Q(s_t, a_t, s_g)$, $f_\phi$, and the contents of $\mathcal{M}$. Iteratively forming plans through PER and executing them creates a feedback loop between these components, where: i) *actions inform perception* during the training of $f_\phi$, ii) *perception facilitates actions* through the formation plans, and iii) *memory serves as the medium* for this reciprocal interaction. As a result, trajectories produced by explicit planning are gradually internalized as implicit behavior encoded in the model parameters. This leads to: Q-values propagating further into distant goals (**Left**), memory contents getting closer to optimal (**Middle**), and performances of local policies showing significant improvement (**Right**).

use SORB [12] and SPTM [14] as baselines.[3] Fig.4 shows the results. In addition to the local policy getting stuck, a new mode of failure for both baselines is that false distance estimates throw-off graph search by setting hallucinated shortcuts. A new baseline is $\pi_{mpc}$, which extends the SPTM local policy by using $p_{fwd}$ and $p_t$ from Sec.2.2 to implement an MPC loop with n-step look-ahead. $\pi_{mpc}$ avoids getting stuck in columns thanks to n-step lookahead, but still isn't sufficient for global navigation as the accuracy of simulated rollouts from $p_{fwd}$ decreases with the number of timesteps.

*Refining Memory Contents:* We refine the contents of $\mathcal{M}$ by iteratively generating and executing $\tau_{\mathcal{M}^*(s_c, s_g)}$. We then retrain all model components only on the resulting new data that is equal in size to the initial unrefined $\mathcal{M}$. Fig.7 shows the results. When $\pi_{\mathcal{M}}$, and $argmax_a Q(a_t, a, s_g)$ are used as policies, their success ratio increases significantly if they are trained on the optimized $\mathcal{M}$. Q-value estimates trained on the optimized $\mathcal{M}$ also propagate better to goals further away. The scaling of $len(\tau_{\mathcal{M}(s_c, s_g)})$ with goal-distance changes from an exponential trend to an approximately linear one, due to the inclusion of transitions from successfully executed $\tau_{\mathcal{M}^*(s_c, s_g)}$. These results highlight that refining memory contents improves the quality of future plans.

## 4.2 Experiments in Habitat

As shown in Fig.8, we find that our method allows image-based navigation in this new domain with significantly different visuals and layouts (i.e., real-world apartments), action space (i.e., turn-left, turn-right, go-forward), and state space (i.e., single $256 \times 256$ RGB images with 150 FOV). Perhaps more surprisingly, we find that training $f_\phi$ only on exploration data from a *single* apartment generalizes substantially well to any unseen apartment, which directly allows perceptual experience retrieval and trajectory stitching when provided with a corresponding replay buffer. For a quantitative evaluation, we randomly pick two apartments, named Roxbox and Annawan. In both apartments, we collect an exploration dataset using a uniform random walk sequence of only 150k timesteps. We

---

[3]For a comparison without confounders, we train SoRB with DDQN [42] rather than distributional Q-learning [43], and we do not employ temporally consistent localization for SPTM, as such fixes are equally applicable to our method and orthogonal to the discussion. The supplementary provides further elaboration.

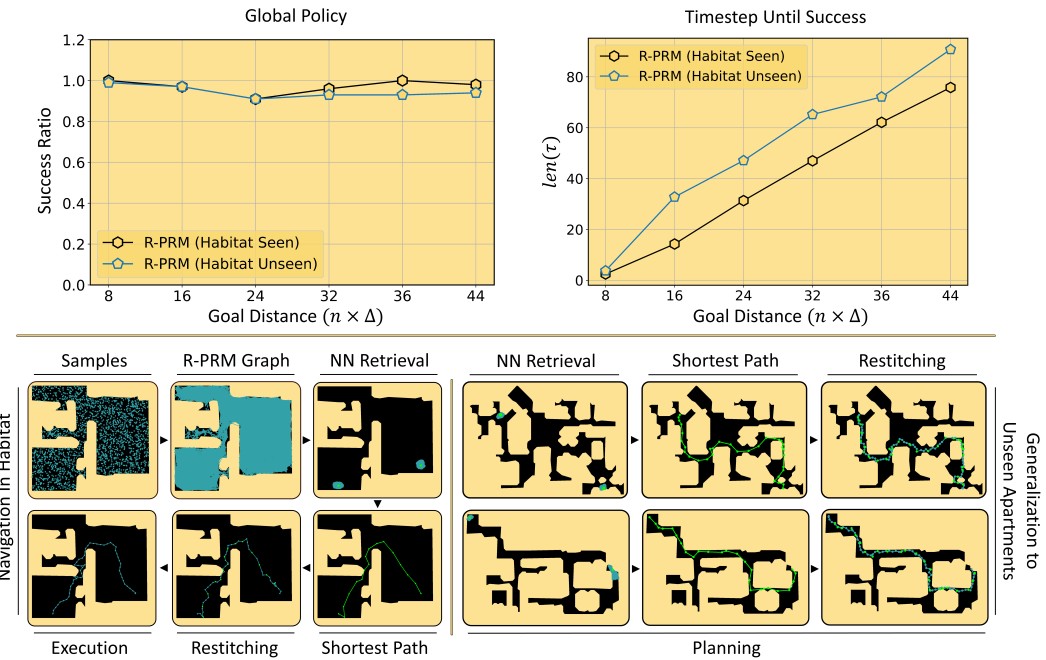

Figure 8: We evaluate our R-PRM based policy $\pi_{\mathcal{M}^*}$ in the Habitat simulator for image-based navigation. **Top Left:** Success ratios in training and test apartments. **Top Right:** Number of timesteps until reaching the goal. ("Habitat seen" refers to the training apartment Roxbox, while "habitat unseen" refers to the test apartment Annawan. **Bottom:** We found that training the perception model $f_\phi$ on a *single* apartment generalizes sufficiently well to allow perceptual experience retrieval and trajectory stitching in any unseen apartment.

train the model components solely on data from Roxbox. We then use them to implement our $\pi_{\mathcal{M}^*}$ policy from Sec.4.1, which we then evaluate on both apartments. For $n \in \{8, 16, 24, 32, 36, 44\}$, we randomly sample 100 pairs of start and goal-states in a way that the geodesic distance between them lies within $n \times \Delta$ and $(n + 8) \times \Delta$ through rejection sampling. A policy is considered successful if it can get within $2 \times \Delta$ proximity of the goal-state. We do not plot the SPTM and SORB baselines, because we found that the models $\pi_{inv}(a|s_t, s_g)$ and $argmax_a\ Q(s_t, a, s_g)$ that they use as local navigation policies achieved almost zero percent success rate in reaching local goals beyond $\sim 2 \times \Delta$ distance. We empirically observed that most of the time these policies get stuck in repetitive rotational motions without moving forward. This is most likely due to the difficulty of offline RL training with hindsight relabelling over random-walk data obtained with a much more challenging non-cartesian action space $\{turn\_left\_30\_deg, turn\_right\_30\_deg, move\_forward\_\Delta\}$.

## 4.3 Experiments in Maze2D

|  | SAC | SAC-off | BEAR | AWR | BCQ | CQL | IQL | Diffuser | PALMER |
|---|---|---|---|---|---|---|---|---|---|
| maze2d-umaze | 110.4 | 145.6 | 28.6 | 25.2 | 41.5 | 31.7 | 89.6 | **182.1** | 131.76 |
| maze2d-medium | 69.5 | 82.0 | 89.8 | 33.2 | 35.0 | 26.4 | 105.2 | 332.9 | **416.28** |
| maze2d-large | 14.1 | 1.5 | 19.0 | 70.1 | 23.2 | 40 | 159.9 | 328.1 | **361** |

Table 1: Total rewards on the Maze2D benchmark, which is a continuous control task that requires long-horizon planning. Our R-PRM based $\pi_{\mathcal{M}^*}$ policy achieves comparatively strong performance.

To test our method on a continuous control task, we perform additional experiments on the Maze2D benchmark. As shown in Table.1, we find that the same $\pi_{\mathcal{M}^*}$ policy from sections 4.1 and 4.2 achieves strong performance, and can solve mazes of all three complexities.

## 5 Discussion and Future Directions

*Is PALMER less expressive than standard deep Q-learning:* Two important premises of deep Q-learning [44, 34] are: i) minimizing Bellman error through temporal-difference (TD) updates can restitch observed transitions in new optimal ways [41, 45], ii) a neural network can learn to extrapolate

Q-values to unobserved but close-by states in high-dimensional spaces (e.g. images) [46]. Both arguments are equally valid for our approach, since it can: i) restitch transitions at arbitrary resolutions (i.e., anywhere from one-step transitions to multi-step trajectories) by virtue of sampling-based planning, ii) group together close-by states through $d_\phi$. Therefore, PALMER is an RL algorithm that: **i)** optimizes Bellman error through sampling-based optimal planning rather than gradient-based TD-updates [46], **ii)** performs extrapolation between states using a perceptual-backbone $f_\phi$ rather than a deep Q-network, and **iii)** replaces the greedy-policy $argmax_a \ Q(s_t, a, s_g)$ and value estimate $max_a \ Q(s_t, a, s_g)$ with $argmax_a \ Q(s_t, a, \tau_{\mathcal{M}^*(s_t,s_g),s,1})$ and $\mathcal{R}(\tau_{\mathcal{M}^*(s_c,s_g)})$ respectively. The key benefits of these alterations come into play when $s_t$ and $s_g$ are far apart, and these benefits are: **i)** the PER mechanism in eq.1 that prevents hallucinations in $Q(s_t, a, s_g)$, **ii)** global propagation of value estimates by virtue of employing sampling-based planning methods, which are known to be particularly proficient at searching high-dimensional state spaces across long-horizons [35, 29].

*Combining PALMER with standard deep Q-learning:* Our approach can also be flexibly combined with any traditional Q-learning method [46, 47, 48], by using our proposed planning algorithms (Sec.2.4) as experience replay methods [49]. This alternative approach stitches together $\tau_{\mathcal{M}^*(s_c,s_g)}$ during training, and perform backwards TD-updates over this trajectory starting from $s_g = \tau_{\mathcal{M}^*(s_c,s_g),s,-1}$ and ending at $s_c = \tau_{\mathcal{M}^*(s_c,s_g),s,0}$. As suggested by Fig.7, this can allow value estimates $Q(s_t, a, s_g)$ to propagate more globally. Our proof-of-concept experiments identify this as a promising direction, and we leave a further extensive evaluation to future work.

*Connections to contingency learning:* Contingency learning refers to the acquisition of knowledge of statistical correlations between percepts [50, 3, 51]. Following this definition, *PALMER can be interpreted as a contingency learning framework, as the latent distance metric $d_\phi$ captures statistically how likely two states are to be observed in close temporal proximity*. The knowledge of these statistical contingencies between states is then used for long-horizon decision making through the proposed perceptual experience retrieval and planning mechanisms.

## 6  Conclusion and Limitations

We presented PALMER, a long-horizon planning method that combines learning-based perceptual representations with classical sampling-based planning algorithms. Given a goal state $s_g$ and reward function $\mathcal{R}$, our method searches the contents of an offline replay-buffer $\mathcal{M}$ to stitch together a sequence of transitions $\tau_{\mathcal{M}^*(s_c,s_g)} = \{s_1, a_1, s_2, ...\}$ that reaches $s_g$ while maximizing $\mathcal{R}$. This results in an experiential framework for long-horizon planning that is significantly more robust and sample efficient compared to baselines.

Our experiments show that PALMER can successfully solve long-horizon planning tasks from continuous high-dimensional inputs. In particular, we have shown that given an offline dataset of only 150k transitions (i.e., compared to sample complexities around the orders of magnitude 1e6-1e7 common in RL) obtained from an entirely uniform random-walk (i.e., which is significantly less structured compared to on-policy rollouts), it allows image-based navigation between any two points in large-scale scans of real-world apartments.

We believe that our memory-based planning perspective highlights a number of interesting questions for future research. First, which transitions should be kept in the replay buffer $\mathcal{M}$, and which ones should be discarded? $\mathcal{M}$ cannot be infinitely expanded after deployment, and it is critical to distill away redundancies between stored experiences. Second, when the environment undergoes a change, which transitions in the replay buffer remain valid and can still be used for planning, and which ones become invalid? A mechanism that can answer this question can allow quick and sample-efficient adaptation to environmental changes. Third, how can we extend $f_\phi$ to allow more abstract associations and functional equivariances between states? This can improve generalization by defining a more flexible notion of experience retrieval that can recycle past behavior in new contexts and for new tasks. We leave these questions to future work.

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
