# OpenReview forum: "PALMER: Perception - Action Loop with Memory for Long-Horizon Planning"
_NeurIPS.cc/2022/Conference — NeurIPS 2022 Accept_

### Official Review · Reviewer_t3oP · 2022-07-08

**Rating:** 7
**Confidence:** 3
**Soundness:** 3 good
**Presentation:** 3 good
**Contribution:** 3 good

**Summary:**

This paper’s main contribution is a method for stitching observed trajectories together to create a graph of the state space on which classical planning can be done to reach a given goal from a given starting point. This method then forms the core of an iterative system that can collect experience given such a graph and then improve the graph (and other learned components) given the new experience. Experiments in VizDoom validate some of the underlying assumptions and choices of the method and demonstrate its benefits over existing work.

**Questions:**

1. How does the proposed method fare as a function of dataset/memory-buffer size? Does performance fall off more quickly or slowly than other methods as the memory buffer shrinks?
2. How strong and how necessary is the assumption of additive reward over trajectories?
3. If I understand correctly, d_\phi[(sc1, sg1), (sc2, sg2)] (from line 74) computes the L2 over concatenated feature vectors, which requires computing a square root over the sum of squared entries. Wouldn’t it be better to compute separate L2 distances over sc1,sc2 and sg1,sg2 respectively and then add those values?
4. Further, the bold text in lines 76-78 says that d_\phi cares about similar actions and costs but d_Q appears to measure the max cost of moving in either direction from sc1<->sc2 or from sg1<->sg2, which says nothing about the actions and costs of the trajectories from sc1->sg1 and sc2->sg2. So, this bolded text seems wrong?
5. How well do SoRB and your method do when using the improvements mentioned in footnote 3? Since it’s possible those improvements would eat into some of the benefits from your method.
6. Please define what an “update” is in figure 7.
7. How well does your method perform in more stochastic domains? I don’t see anything in the method to deal with noisy transitions.


**Limitations:**

Yes, the authors have done a sufficient job addressing limitations and potential negative societal impacts of their work.

**Strengths And Weaknesses:**

**Originality:** The presented work is a natural extension of existing ideas in the literature of goal-conditioned planning but is comprehensive and clearly adds novelty to the existing literature by introducing an alternative state embedding function, supporting arbitrary rewards, and enabling the use of classic planning algorithms on in-memory observations.

**Quality and clarity:** The presentation is of high quality, well-written, and clear, albeit quite dense. My main complaint in this regard is that the authors seem to be quite constrained by the page limit, and this is impacting the readability of parts of the text and definitely of the figure captions. I’d encourage moving an additional paragraph from somewhere into the supplement and clarifying the figure captions (e.g., Fig 3 needs the supplement to be understood at all, which should not be the case).

**Significance:** The contribution appears significant and the evaluation is reasonable but a bit limited in scope (mainly focused on ViZDoom) so it’s unclear what the longer term significance of this work will be.

Overall, the paper is well-written and fairly enjoyable to read (aside from the density and need to refer to the supplement to understand certain sections and figures). The exposition, explanations, and clear connections to existing work are much appreciated. This is a large and somewhat complicated method and I felt each component was explained and justified and connected to the other components.

---

> ### Author Response · Authors · 2022-08-02
> **Response to reviewer t3oP (Part 2)**
>
> **How well would SoRB do with the additional improvements from footnote 3?**
>
> We think that the most significant discrepancy between our implementation of SoRB and the original paper is that we train the Q-function on an offline dataset of transitions rather than employing an online RL setup (as also discussed in lines L257-269 in the supplementary). Since PALMER is intended to be an offline planning method, we opted not to include the online version of SoRB as a baseline.
>
> **Please define what an “update” is in figure 7.**
>
> As mentioned in L224-226 in the original paper, an update in Fig.7 refers to iteratively generating and executing planned trajectories to collect a new dataset of transitions that is equal in size to the original dataset, and then retraining all model components from scratch only on the new dataset. A more elaborate description can also be found in Algorithm 9 from the supplementary. We will improve the figure caption and the explanation in the main body of the text to make this point more clear.
>
>
> **How well does your method perform in more stochastic domains?**
>
> There are two mechanisms that are intended to address stochastic transitions in PALMER:
>
> i) Constraint 5 in the PMR equation (i.e., between L111-112 in the main paper) compares the observed length $len(\tau)$ of a retrieved trajectory with the mean of an estimator $p(T | f_\phi(s_t), f_\phi(s_g))$ of the expected number of timesteps necessary to traverse between its start and goal states. This means that if an observed sequence of transitions $\tau$ is an outlier and has a low probability of happening, constraint 5 will be violated and PMR will not retrieve it for the current/goal state pair $(s_t, s_g)$. This way, PMR only uses trajectories that have a high probability of successful traversal.
>
> ii) One can always run PMR in a receding horizon manner, which creates a close loop controller that can compensate for stochasticity.
>
> The table below gives an evaluation of the success ratio for the policy $\pi_{\mathcal{M}^*}$ from Fig.4, in the same VizDOOM environment only with stochastic transitions. Every original action that moves the agent by a distance $\Delta$ is corrupted by an additional distance $\Delta / 3$ towards a uniformly random direction sampled from the four cardinal directions. The results show that the success rate is robust to stochastic transitions.
>
> | Goal Distance ($n \times \Delta$) | 1 | 2 | 3 | 4 | 5 | 6 | 7 | 8 | 9 | 10 | 11 | 12 | 13 | 14 |
> | --- | --- | --- | --- | --- | --- | --- | --- | --- | --- | --- | --- | --- | --- | --- |
> | Success Ratio (%) | 0.8875 | 0.900667 | 0.90525 | 0.91 | 0.919 | 0.904 | 0.90025 | 0.88975 | 0.901 | 0.897 | 0.911 | 0.915 | 0.926 | 0.909 |

---

> > ### Comment · Reviewer_t3oP · 2022-08-08
> > **Update**
> >
> > Thank you for your comprehensive responses. I'm happy to keep my score at 7.

---

> > > ### Author Response · Authors · 2022-08-08
> > > **Thank you**
> > >
> > > Thank you again for your time spent reviewing our paper. We also sincerely appreciate the constructive suggestions and kind comments.

---

> ### Author Response · Authors · 2022-08-02
> **Response to reviewer t3oP (Part 1)**
>
> Thank you for your suggestions, questions, and kind comments. We are happy to hear our paper was an enjoyable read, and appreciate the positive feedback about it being “comprehensive, novel, and significant”. Below we adress the remaining comments and questions.
>
> **The presentation is dense and occasionally requires referring to the supplementary material.**
>
> We fully acknowledge and appreciate this comment, and we will extensively revise our paper to be a much smoother read.
>
> **The evaluation is reasonable but a bit limited in scope.**
>
> Thank you for the comment. We will revise our paper to better highlight experiments in Habitat, which are currently presented mostly in the supplementary material between L328-373. We also added experiments in the Maze2D benchmark, to show that our method can optimize general reward functions on continious control tasks.
>
> **How strong and how necessary is the assumption of additive reward over trajectories?**
>
> The state-wise additive reward function assumption $\mathcal{R}(\tau)=\sum_\tau r(s_t,a_t,s_{t+1})$ that is almost ubiquitously employed in RL and the property $\mathcal{R}(\tau_1 \circ \tau_2) = \mathcal{R}(\tau_1) + \mathcal{R}(\tau_2)$ we state in our paper are biconditional statements, since every $\tau$ is a concatenation of tuples $(s_t, a_t, s_{t+1})$. We only employ the latter notation because it naturally highlights the fact that PALMER plans by concatenating trajectory segments.
>
> **Wouldn’t it be better to compute separate L2 distances over sc1,sc2 and sg1,sg2 respectively and then add those values?**
>
> Yes, we completely agree. This is in fact how we implement PMR. The notation in L74 is mostly employed for a compact exposition, and we will revise it as it is likely to create confusion.
>
> **Is the the bold text in lines 76-78 wrong?**
>
> Lines L66-81 in the supplementary contain an errata that fixes equations 1 and 2 in the main paper. The same section also contains a derivation of how $2d_Q$ forms an upper bound on the difference between the costs of the two shortest paths sc1 <-> sg1 and sc2 <-> sg2 (together with a figure that visualizes the calculation). This difference between the costs of path sc1 <-> sg1 and path sc2 <-> sg2 essentially intends to capture how similar following the two paths are from an optimal planning perspective (i.e., what is the worst cost possible if the planner substitutes sc1 <-> sg1 with sc2 <-> sg2).

---

### Official Review · Reviewer_Bn9U · 2022-07-10

**Rating:** 6
**Confidence:** 4
**Soundness:** 3 good
**Presentation:** 3 good
**Contribution:** 3 good

**Summary:**

This paper proposes a method similar to anytime-sampling methods in planning (e.g. RRT*) but in the RL context, to stitch together trajectories from an experience buffer to solve goal reaching tasks.

The core idea is to learn representations using general neural function approximators by leveraging the idea that the functional distance between two interactions should be small if they can be achieved through similar actions and attain similar reward/cost values. This is formalized using state value function and L2 function over learn feature embeddings.

The representation is learn by fitting several sub-modules including an encoder, a trandition dynamics model, action proposal network and a function to retrieve a distribution over timesteps necessary to achieve an interaction. Effectively this gives rise to a policy improvement algorithm that follows a virtuous loop of improve the system to achieve goal by iterating over: 1) learn an accurate Q function, 2) Use this Q function learn better function appx, 3) Use this to better estimate expected reward given start and goal query, 4) this is turns lead to better exploration, repeat. This is the core algorithm and there are parallels drawn to RRT* algorithms which behave in similar ways but by using hand-crafted sub-functions. This reformulation of planning algorithms as a memory retrieval mechanism is interesting and is the core conceptual contribution of this paper. This is then validated on navigation tasks in VizDoom and Habitat.

**Questions:**

-  It looks like the state action value function has not been trained in the Habitat environment fully correctly. There are a lot of glitches that happen in the policy rollout. Why is this?

- "We say two states approximately overlap if dϕ(st, sg) ≤ dp holds, where dp is a hyper-parameter". How is the hyper param selected? It seems like this would greatly depend on the topology of the environment. The variable would be vastly different for navigation, manipulation or gaming tasks. This also makes me wonder if the algorithm easily generalizes to environments beyond navigation tasks. In principle it should but the exploration and this assumption can make it highly trivial to get good empirical results. Have the authors ran experiments on other tasks like the Mujoco continuous control suite or Atari? Even experiments in these synthetic domains would prove out the empirical generality of this framework and make it a very strong paper.

- Section 2.3 is not clear at all. How is the search done?  "Given any interaction proposal (sc, sg) as a query, we use the components from Sec.2.2 to retrieve the highest scoring trajectory from M whose first-last states overlap with the query pair according to our approximate-overlap criterion from Sec.2.1.". More details would be good.

- If the environment is dynamic, how would this approach still work?

**Limitations:**

- RRT* style of algorithms come with anytime sampling convergence guarantees. What does and dosen't hold in this setting?

- Are there any situations under which the bellman optimality is broken under this policy improvement scheme?

**Strengths And Weaknesses:**

Strengths:

In long horizon tasks like navigation under sparse rewards, exploration and policy improvement is challenging. Assuming that the agent explores the state space well enough, the experience stitching approach proposed in this paper is a principled solution to tackle this problem in case of non-linear function approximation based RL and planning.

The key empirical result is shown in Figure 4 where the success ratio degrades more gracefully than baselines in the long horizon planning setting.

Weakness:

This method assumes that the exploration problem is "solved". One of the advantages of RRT* like algorithms is that they quickly explore large unstructured spaces. However, a trajectory stitching based approach will only work well assuming the space is well explored. Please discuss limitations.

The paper would also be much more powerful and the parallels to RRT* like algorithms much more grounded if larger environments were considered. I suspect some form of intrinsic motivation or curiosity would be needed to solve this but it is important to know where the current proposed method breaks (e.g. what maze or level complexity would this break on compared to RRT*, assuming knowledge of state space?).

---

> ### Author Response · Authors · 2022-08-02
> **Response to reviewer Bn9U (Part 2)**
>
> **Does PALMER work for continious control tasks? Such demonstrations would make this a very strong paper.**
>
> Thanks for the suggestion. We evaluated PALMER on the Maze2D continious control benchmark, and posted the results as part of the general response above.
>
> **How exactly is the search in PMR done?**
>
> L159-168 in the supplementary (i.e., the section titled “How do we solve the optimization problem from equations 3,4,5 from the main paper?”) provides an elaborate discussion.
>
> **If the environment is dynamic, how would this approach still work?**
>
> This is an excellent and relevant question. We assume that in this context a “dynamic environment” refers to an environment where the state-transition dynamics change over time (e.g., certain transitions in the memory buffer become infeasible over time). As mentioned in L275-278 in the main paper, this is an exciting direction that we are currently working on. We generally need a mechanism that can identify transitions that become invalid, so that they aren’t used for planning anymore. For the current version of PALMER, a trivial approach would be to remove an edge from the planning graph whenever its traversal fails more than a given number of times (as well as removing the corresponding transitions from the replay-buffer $\mathcal{D}$). We note that for example, a Q-learning approach would also adapt in a similar manner, by attempting and failing to traverse the same invalid state-transition a number of times until the new Q-values propagate and adjust accordingly through TD updates. The default Q-learning algorithm has no control over when TD updates can eventually cause such a change, and we therefore think that an explicit graph and count-based strategy to keep track of invalid transitions can achieve quicker adaptation.
>
> **Does PALMER provide any theoretical guarantees?**
>
> The convergence guarantees for RRT/PRM make explicit use of the geometric structure underlying the kinematic motion-planning problem. Since PALMER contains a deep-learning based state embedding function and can only plan over previously seen state transitions, we do not provide any convergence bounds (i.e., as $s_g$ may lie out of $\mathcal{D}$, or the learning based $d_\phi$ may not perfectly correlate with real local-reachability). Similarly, theoretical arguments about the bellman optimality of RL algorithms usually assume either discrete state-action spaces, or hold at the limit where every state action pair is sampled an infinite number of times. Both of these are not applicable in our setting as we plan over a fixed offline dataset of images.

---

> ### Author Response · Authors · 2022-08-02
> **Response to reviewer Bn9U (Part 1)**
>
> Thanks for your suggestions, questions, and for finding our paper “interesting” and “principled”. We address the feedback given under the weaknesses, questions, and limitations sections below:
>
> **This method assumes that the exploration problem is "solved”. Please discuss limitations.**
>
> As we briefly suggest in L179-183 of the main paper, and discuss in the general comments posted above, our method assumes an offline dataset $\mathcal{D}$ of state-space transitions as given. This is directly analogous to the assumptions made by RRT/PRM that a geometric environment model as well as a collision checking module are available. These two assumptions in RRT/PRM and $\mathcal{D}$ in PALMER serve the exact same function: defining the set of all feasible state space transitions available for search and planning.
>
> To elaborate, a geometric model $\mathcal{C}$ can be thought of as the set of all available line segments that can be traversed in a valid way (i.e., without collision), and RRT/PRM then operate on this set by stitching together such line segments. What RRT/PRM doesn't do is to expand the boundaries of $\mathcal{C}$ to include new line segments. When we say RRT* like algorithms quickly "explore" large unstructured spaces in a *given* model $\mathcal{C}$, what we mean is they search *within* $\mathcal{C}$ and quickly build paths that connect its contents.
>
> Exactly in the same way, the memory buffer $\mathcal{D}$ in our paper is the set of all available state transitions that were observed to be traversable, and PALMER operates on this set by stitching together such transitions to quickly build paths that connect its contents. Much like how RRT or PRM plan strictly within $\mathcal{C}$ and do not aim to expand (i.e., explore beyond) its contents, our search and planning algorithm also plans strictly within $\mathcal{D}$ and does not aim to expand its contents.
>
> **The paper would also be much more powerful if larger environments were considered.**
>
> Our paper contains demonstrations on two large-scale scans of real-world apartments: i) Roxboro, with a total area of ~62 m2, and ii) Annawan, which has a total-area of ~75 m2. As mentioned in L352-373 of the supplementary, given an offline dataset of only 150k transitions (i.e., compared to sample complexities around the orders of magnitude 1e6-1e7 common in RL) obtained from a uniform random-walk (i.e., which is significantly less structured compared to on-policy rollouts), PALMER allows image-based navigation between any two points in both environments. We also provided results on Maze2D in the rebuttal.
>
> **At which maze or level complexity would PALMER break?**
>
> We think that whether PALMER breaks or not depends more on how well the dataset $\mathcal{D}$ covers the environment, which is assumed to be given. PALMER doesn’t constitute an exploration process on its own, and as long as $\mathcal{D}$ provides sufficient coverage PALMER can be employed to search its contents.
>
> **Why does the policy glitch in Habitat?**
>
> L352-L373 in the supplementary (i.e., the section titled “Why does the agent occasionally take random-looking actions in the habitat navigation trials”) intends to provide an elaborate discussion on this, as well as potential ways to counter it.
>
> **How is the hyperparameter $d_p$ selected?**
>
> As mentioned in L143-157 from the supplementary, $d_p$ is picked by examining the average  $d_\phi$ distance between subsequent states in the memory-buffer.

---

### Official Review · Reviewer_BFRQ · 2022-07-11

**Rating:** 7
**Confidence:** 4
**Soundness:** 3 good
**Presentation:** 4 excellent
**Contribution:** 2 fair

**Summary:**

In this work, the authors introduce PALMER that combines reinforcement learning and classical planning (particularly sampling-based methods) to achieve robust navigation in visual domains. PALMER maintains a memory buffer. During rollout, given a state-goal pair, the lowest cost trajectory on the graph (w.r.t state-goal query) is found by solving an optimization problem minimizing a similarity metric based on perceptual representations (processed referred to as perceptual memory retrieval, or PMR). A probabilistic roadmap is incrementally constructed with transitions added to the buffer. This is done by following the classical sampling-based motion planning routines but replacing the Euclidean distance metric with one that's based on PMR. Doing so allows for effective use of the memory buffer as a trajectory bank for high-level guidance of long horizon navigation tasks. Simulation results show that PALMER is able to effectively generate goal-reaching trajectories from visual input and exhibit decent generalization capability in the VizDoom and Habitat environments

**Questions:**

1. The claims in the paper are that PALMER is more robust to false predictions and it learns abstractions that can extend from goal-reaching to general RL problems. However, it is not clear how these two claims are supported by the experiment results. In order to support these two claims, the authors need to show (a) how the number of false predictions correlate to the task success rate in PALMER and comparison methods and (b) how PALMER performs on tasks more complex than visual navigation.

2. The idea of restitching transitions in the memory buffer is relevant to offline RL (A Survey on Offline Reinforcement Learning: Taxonomy, Review, and Open Problems ), can the authors comment on the similarities and differences?

3. Out of all the components in PALMER, which is the most time consuming and how does it scale with the dimension of its inputs?

4. The effectiveness of PMR and the constructed roadmap depends on coverage of the transitions in the memory buffer over the task space. Will this hinder exploration at the start of training when the coverage is low?

**Limitations:**

requiring a graph-based roadmap inherits some of its limitations, including the inefficiency caused by high dimensional state space and closely situated obstacles a cluttered environment. And it's not clear to me how a sampling-based planner can be extended to non-navigation RL tasks with arbitrary reward.

**Strengths And Weaknesses:**

Strengths
1. The paper is well written, and the visualizations are very helpful in explaining the core ideas.
2. I appreciate the detailed explanations of the key concepts provided in the paper and appendices, again incredibly helpful for understanding the paper.

Weaknesses
1. lacks comparison with other planning and RL methods in visual navigation
2. evaluation in support of the claims would strengthen the paper

---

> ### Author Response · Authors · 2022-08-02
> **Response to reviewer BFRQ (Part 2)**
>
> **Restitching transitions in the memory buffer is relevant to offline RL.**
>
> We definitely agree with this observation. While PALMER by itself only constitutes a planning method and not a full offline RL agent, it can be turned into one by combining it with a local policy $\pi(a \ | \ s_t, \, \tau_{plan})$, as elaborated in the general response above. L244-265 in the main paper also present a related discussion about how PALMER can be interpreted as an RL method.
>
> **Computation time characterization of PALMER.**
>
> Thanks for this suggestion. The table below shows how the computation time (in seconds) of every step in PALMER scales with graph size. We particularly note that solving the PMR optimization is faster than neural network inference. Building the planning graph takes the most time, but is a one time only operation.
>
> | Graph Size | $f_\phi$ forward pass | PMR Solution | Node Sampling | Graph Building (PRM) | Planning |
> | --- | --- | --- | --- | --- | --- |
> | 250 | 1.095e-3 | 8.41e-4 | 2.615e-1 | 7.313 | 9.084e-3 |
> | 500 | 1.095e-3 | 8.41e-4 | 4.644e-1 | 25.334 | 2.817e-2 |
> | 750 | 1.095e-3 | 8.41e-4 | 6.333e-1 | 56.368 | 1.971e-2 |
> | 1000 | 1.095e-3 | 8.41e-4 | 1.009e-1 | 108.494 | 1.739e-2 |
>
> **The effectiveness of PALMER depends on the coverage of the memory buffer over the task space. Will this be a bottleneck?**
>
> We note that PALMER doesn’t address exploration on its own, and assumes that an offline dataset of transitions is given. The discussions in section 2.5 (i.e., “Closing the Perception-Action Loop to Optimize Memory Contents”) as well as the corresponding experiments are only intended as a demonstration that PALMER can be used to inform and accelerate a given downstream auxiliary exploration process. Therefore for any task, whether exploration will be a bottleneck or not depends on the particular exploration method PALMER is combined with.
>
> **High-dimensional state spaces present a problem for sampling based planning, is this also the case for PALMER?**
>
> As discussed in Sections 2.1 - 2.2 of the main paper, one contribution of our method is a state embedding function that projects a high-dimensional state space into a low dimensional one where the metric correlates with reachability. Our image-based navigation experiments in turn demonstrate that sampling-based planning in PALMER is tractable on state spaces that have $256 \times 256 = 65536$ dimensions.

---

> > ### Comment · Reviewer_BFRQ · 2022-08-07
> > **Response**
> >
> > I thank the authors for addressing my questions. The added results and explanations help strength the arguments of the paper. Upgraded score to 7.

---

> > > ### Author Response · Authors · 2022-08-08
> > > **Thank you**
> > >
> > > We are happy to hear that the additions were adequate. Thank you for your time spent reviewing our paper, as well as for the helpful suggestions that improved it.

---

> ### Author Response · Authors · 2022-08-02
> **Response to reviewer BFRQ (Part 1)**
>
> Thank you for your suggested improvements on the paper, and we are glad to hear that the appendix and the visualizations proved to be useful. We hope the following adequately addresses the comments and questions raised under the weaknesses, questions, and limitations sections:
>
> **The method lacks comparison with other planning and RL methods in visual navigation.**
>
> As we also mention in the general response above, the main motivation behind our method is to design a flexible planning interface that is compatible with any local controller and any auxiliary data-collection/exploration mechanism. As such, it is important for our method as well as the relevant baselines to operate strictly on an offline dataset of unlabeled transitions that solely consist of high-dimensional on-board sensory data (e.g. images), without assuming any auxiliary instrumentation in the environment or oracle information that cannot be sensed by the agent. Most planning and RL methods and benchmarks make assumptions that do not fit into these requirements, and as such we picked our baselines accordingly and opted to implement our own evaluation environments in VizDoom and Habitat (i.e., rather than evaluating on standard embodied navigation benchmarks like habitat pointnav).
>
> **How do the experimental results show that PALMER is more robust? The authors should show how the number of false predictions correlate to the task success rate.**
>
> Thanks a lot for the suggestion. Please find below a table of how the number of false edges change with the graph size (together with the number of correct edges, and the total number of edges set by each method). This table, together with Fig.4 of the main paper shows that PALMER is robust, doesn't set false edges, and as a result achieves a high success rate.
>
> | Graph Size | Correct Total | PMR Total | PMR Faulty (%) | SPTM Total | SPTM Faulty (%) | SoRB Total | SoRB Faulty (%) |
> | --- | --- | --- | --- | --- | --- | --- | --- |
> | 250 | 10882 | 3120 | 0 | 4966 | 2 | 1703 | 13 |
> | 500 | 44064 | 13155 | 0 | 20273 | 2 | 6347 | 10 |
> | 750 | 98998 | 29659 | 0 | 45551 | 2 | 13540 | 8 |
> | 1000 | 175966 | 52613 | 0 | 81037 | 2 | 23874 | 7 |
> | 1500 | 392122 | 11062 | 0 | 179610 | 2 | 50854 | 5 |
>
> **How can a sampling-based planner be extended to non-navigation tasks with arbitrary reward? The authors should show demonstrations.**
>
> The result of running PALMER is an RRT or PRM type of planning graph where each edge contains both the total number of timesteps required for its traversal as well as the total reward obtained by doing so. Given this data-structure the following are possible:
>
> i) Given a current node and goal node, find all simple paths reaching the goal node and follow the path with highest reward.
>
> ii) Given only a current node, find shortest time paths to all target nodes (e.g., using Floyd’s algorithm), and follow the path with highest reward.
>
> The general response posted above includes a demonstration of the second method on the Maze2D benchmark.

---

### Official Review · Reviewer_PChv · 2022-07-13

**Rating:** 6
**Confidence:** 4
**Soundness:** 3 good
**Presentation:** 2 fair
**Contribution:** 2 fair

**Summary:**

This paper presents a planning method called PALMER that creates a feedback loop between reinforcement learning, representation learning, sampling-based motion planning and a non-parametric memory. The method tackles a specific variant of a planning problem where observations from a prior exploration of the environment are available and the goal is to find optimal paths between any two states in the  same environment. The authors conduct experiments in the ViZDoom and Habitat environments which show some improvements over prior methods.

**Questions:**

See questions in weaknesses above.

**Limitations:**

Yes

**Strengths And Weaknesses:**

Strengths:
- The method is well motivated. The idea of using sampling-based motion planning for this task makes sense. The use neural networks for learning the graph structure required for motion planning from pixel-based observations is clever.
- The results on ViZDoom environment show a significant performance improvement especially for distant goals.

Weaknesses:
- The scope of the method is low in my opinion as the method requires access to an exploration of the environment.
- Since the authors are tackling embodied tasks, there are alternative methods which build a map of the environment and use differentiable planning, for example Gated Path Planning Networks (Lee et al. ICML 2018), Path Planning using Neural A* Search (Yonetani et al. ICML 2021), Learning to Plan in High Dimensions via Neural Exploration-Exploitation Trees (Chen et al. ICLR 2020). It is difficult to understand the significance of the proposed approach as compared to this line of work as it is missing in the paper both in related work and as baselines.
- The results on Habitat environment are only qualitative and are not compared against baselines. More comprehensive experiments in realistic environments would make the submission stronger.
- The writing clarity can be improved significantly.
  - Many important experimental details are missing which makes it difficult to assess the significance of the results. It is unclear how the exploration trajectories are obtained in different environments. How many trajectories do you use per environment? How are the trajectories sample, is it a random policy, a trained policy or human trajectories? How are the target images sampled? Are target images always observed in exploration trajectories or can they be novel images?
  - The text contains several terms which are not clearly defined. For example, what do you mean by "approximate-overlap criteria (L37, L50), what are "actually observed transitions" in L48? Which transitions are observed but not actually observed? The text also contains very long sentences which are difficult to follow (for example L39-42, L42-45).

---

> ### Author Response · Authors · 2022-08-02
> **Response to reviewer PChv (Part 2)**
>
> **Results on Habitat are qualitative, experiments on realistic environments would make the paper stronger.**
>
> Part of this was actually provided in the supplementary (which we will clarify in the camera ready). Our Habitat experiments contain demonstrations on two large-scale scans of real-world apartments: i) Roxboro, with a total area of ~62 m2, and ii) Annawan, which has a total-area of ~75 m2. As mentioned in L352-373 of the supplementary, given an offline dataset of only 150k transitions (i.e., compared to sample complexities around the orders of magnitude 1e6-1e7 common in RL) obtained from a uniform random-walk (i.e., which is significantly less structured compared to on-policy rollouts), PALMER allows image-based navigation between any two points in both environments.
>
> Habitat evaluations in fact do report quantitative values for the success ratios and timesteps until success, as shown in Fig.8 of the supplementary. As further explained in lines L345-351 in the supplementary, success ratios for the local and global policies of SPTM and SoRB baselines were almost %0 in Habitat (as reported in the table below), so they weren’t explicitly shown in Fig.8.
> | Goal Distance ($n \times \Delta$) | 8 | 16 | 24 | 32 | 36 | 44 |
> | --- | --- | --- | --- | --- | --- | --- |
> | SPTM Success Ratio (%) | 0.28 | 0.01 | 0.0 | 0.0 | 0.0 | 0.0 |
> | SoRB Success Ratio (%)  | 0.42 | 0.0 | 0.0 | 0.0 | 0.0 | 0.0 |
> | PALMER Success Ratio (%) | 0.99 | 0.97| 0.91| 0.93| 0.93| 0.94
>
> **The writing clarity can be improved, important experimental details are missing.**
>
> We appreciate this comment, and acknowledge that the current version of our exposition is quite dense. We will revise the paper to make it an overall much smoother read. In particular, a large part of the details were provided in the supplementary, which we will address in the final version. As mentioned in lines L55-57 in the main paper, and further elaborated on in lines L280-282 and L332-334 in the supplementary, exploration trajectories in VizDOOM and Habitat were obtained from uniform random-walk sequences of length 300k and 150k time steps respectively, using their corresponding action spaces, and without any rewards or resets. Similarly, as mentioned in lines L202-204 of the main paper as well as lines L301-304 and L339-351 in the supplementary, the start and target images used in the evaluations are sampled from a uniform random distribution over empty space in both environments. As such, they are novel images that were not necessarily observed in the exploration trajectories. Lines L83-L93 in the supplementary (i.e., the section titled “What does approximate-overlap criterion mean?”) elaborates on what the term approximate-overlap criterion means. Also, we agree that all observations are by definition actual, and the redundancy was mostly intended for emphasis.

---

> ### Author Response · Authors · 2022-08-02
> **Response to reviewer PChv (Part 1)**
>
> Thanks for the constructive feedback. We also appreciate the positive reinforcement about our approach being “well-motivated”, “clever”, with “significant performance improvements”. Below, we address the questions and concerns with new experiemental results and clarifying discussions:
>
>
> **The proposed method assumes access to a previously collected dataset of interactions, which limits its scope.**
>
> This is true, but an assumption widely made by several subfields. The assumption of an existing characterization of the environment (e.g., a geometric model, or a previously collected dataset of interactions) is not specific to our method, but rather a common assumption made by all kinematic motion planning algorithms (e.g., RRT, PRM), as well as all offline RL algorithms [[1](https://arxiv.org/abs/2005.01643v2)][[2](https://arxiv.org/abs/2004.07219)][[3](https://arxiv.org/abs/2109.10813)]. We shortly discussed why we think this assumption is valid in the general discussion posted above (i.e., the section titled “i) Why assume $\mathcal{D}$ is given”). For a more elaborate discussion about why separating data-collection (i.e., exploration) from the downstream learning of decision making and planning might be a good idea, we point to a highly-cited offline RL review article [[1](https://arxiv.org/abs/2005.01643v2)].
>
>
> **Methods such as “Gated Path Planning Networks”, “Path Planning using Neural A\* Search”, and “Learning to Plan in High Dimensions via Neural Exploration-Exploitation Trees” are relevant baselines that are missing from the paper.**
>
> As mentioned in lines L1-2, L54-55, L182-183, L267-268 from our main paper, as well as the general discussion posted above, the core premise of our method is that it is flexible enough to operate on an offline dataset of unlabeled transitions that solely consist of high-dimensional on-board sensory data (e.g. images), without assuming any auxiliary instrumentation in the environment or oracle information that cannot be sensed by the agent (such as a top-down map or a geometric model). The referenced papers each make a number of assumptions that violate this premise. For instance, they commonly assume an euclidean map-based representation.
>
> In particular, ‘Neural A* Search’ takes as input a *top-down 2D map on which the start and goal positions are marked*. It is also trained in a supervised way on a *labeled* dataset (i.e., 2D maps with ground truth occupancy annotations). ‘Gated Path Planning Networks’ is based on value-iteration networks [[4](https://arxiv.org/abs/1602.02867?context=cs)], and it operates either on an $n \times m$ top-down map, or an equivalent representation based on an $n\times m$ grid of panoramic images. This method is also trained in an *online* way using an *oracle reward function*. ‘Neural Exploration-Exploitation Trees’ assumes a workspace map and a collision checking utility, which are the same assumptions made by classical sampling based planners that our method intends to relax.

---

### Author Response · Authors · 2022-08-02
**General Response (Part 2)**

## 2) Additional Experiments


We tested PALMER on Maze2D environments from the D4RL benchmark [1], and included sample videos in the supplementary material. The results provided in the table below suggest PALMER is a general yet strongly competitive method. In this demonstration, PALMER searches the offline memory buffer with R-PRM at every time-step to produce $\tau_{plan}$, and the downstream controller $\pi(a \ | \ s_t ,\ \tau_{plan} )$ is implemented by simply executing the first action in $\tau_{plan}$, thus forming a receding horizon control loop. We note two things about this control loop:

i) PALMER doesn’t use any additional information that isn’t available to the baselines (i.e., it is still a fully experiental method).

ii) While it works very well for the Maze2D environment, it doesn’t produce a reliable controller $\pi$ on tasks with more complex dynamics (for reasons discussed below).

|          | maze2d-umaze | maze2d-medium | maze2d-large |
|----------|--------------|---------------|--------------|
| BC       | 29.0         | 93.2          | 20.1         |
| SAC      | 110.4        | 69.5          | 14.1         |
| SAC-off  | 145.6        | 82.0          | 1.5          |
| BEAR     | 28.6         | 89.8          | 19.0         |
| BRAC-p   | 30.4         | 98.8          | 34.5         |
| BRAC-v   | 1.7          | 102.4         | 115.2        |
| AWR      | 25.2         | 33.2          | 70.1         |
| BCQ      | 41.5         | 35.0          | 23.2         |
| aDICE    | 2.2          | 39.6          | 6.5          |
| CQL      | 31.7         | 26.4          | 40           |
| IQL      | 47.4         | 34.9          | 58.6         |
| Diffuser | 113.9        | 121.5         | 123.0        |
| PALMER   | 131.76       | 416.28        | 361          |

Table 1. Performance of PALMER with 12 offline RL baselines on the Maze2D environments from the D4RL benchmark [1]. Metrics for the first 10 baselines are taken from [1]. The results for IQL and Diffuser are taken from [2].

For other continious control tasks with more complex dynamics from the same benchmark, we couldn’t find or train a low-level controller $\pi(a \ | \ s_t, \tau_{plan})$ that could reliably track the planned trajectories $\tau_{plan}$ produced by PALMER (e.g., training a goal conditioned Q-value function and using $argmax_a \, Q(s_t,a,s_g)$ as a controller didn’t work). Since PALMER assumes a sufficiently powerful controller $\pi$ that can reliably track $\tau_{plan}$ as a given, we didn't manage to produce additional demonstrations on more continious control tasks besides Maze2D in the rebuttal time. We will add such demonstrations in the camera ready when we manage to find or train an adequate controller.

We thank all reviewers again for their detailed suggestions and constructive criticism, which will definitely improve our paper. We will revise the camera ready according to the reviews and the responses provided here.

[1] Fu, Justin, et al. "D4rl: Datasets for deep data-driven reinforcement learning." arXiv preprint arXiv:2004.07219 (2020).

[2] Janner, M, et al. "Planning with Diffusion for Flexible Behavior Synthesis." arXiv preprint arXiv:2205.09991 (2022).

---

### Author Response · Authors · 2022-08-02
**General Response (Part 1)**

We thank all reviewers for their time and valuable feedback. We are happy they generally found our paper to be “clever with significant improvements”, “principled”, and “clearly novel”.

We noticed two main suggestions made by all reviewers:

1. the exposition is dense and requires referring to the supplementary material to fully understand the method.
2. the paper would be much stronger if it included demonstrations other than image-based navigation.

We address these general suggestions as follows:

1. We provide an additional discussion below that intends to communicate more clearly the main motivations, assumptions, contributions, and limitations of PALMER. The paper will be revised to include this discussion for the camera ready.
2. We provide additional experiments on the Maze2D benchmark, which intend to demonstrate that PALMER can be applied to continious control tasks with sparse and general reward functions.


## 1) Additional Discussion and Clarification

**What is PALMER**: It is a planning algorithm that sits as an interface between any given local controller $\pi$ and any given offline dataset $\mathcal{D}$ of state transitions. Given a goal state $s_g$ and reward function $\mathcal{R}$, it searches the contents of $\mathcal{D}$ to stitch together a sequence of transitions $\tau_{plan} = \{s_1,a_1,s_2,...\}$ that reaches $s_g$ while maximizing $\mathcal{R}$. This sequence is communicated to $\pi$ to be executed, which then communicates back the resulting real transitions to update the contents of $\mathcal{D}$.

**Assumptions**:  PALMER assumes that the following are given:

i) a local controller $\pi(a \ | \ s_t ,\ \tau_{plan} )$, which converts $s_t$ and $\tau_{plan}$ to actions. $\pi$  can be an RL policy, a PID controller, a trajectory optimizer, or any other method. This is conceptually equivalent to the assumption of a downstream control pipeline made by RRT/PRM: $\tau_{plan}$ is analogous to a kinematic trajectory, and $\pi$ is analogous to the downstream control pipeline that executes it (e.g., PID, LQR, trajectory optimization).

ii) an offline dataset (i.e., memory buffer) $\mathcal{D}$, which is conceptually equivalent to the assumption of a geometric model (e.g., mesh, voxel grid, signed distance field) made by RRT/PRM. Much like how a geometric model defines the set of all line segments that can be traversed in a valid way (i.e., without collisions), $\mathcal{D}$ defines the set of all state transitions that were observed to be traversable by the agent. Therefore, both a geometric model in RRT/PRM and $\mathcal{D}$ in PALMER serve the exact same purpose: defining the set of all feasible state space transitions available for search and planning.

**Contributions**: PALMER is intended to be an “experiential planning” method that operates on an offline dataset of unlabeled transitions that solely consists of high-dimensional on-board sensory data (e.g. images), ***without assuming any auxiliary instrumentation in the environment or oracle information that cannot be sensed by the agent on its own.*** The goal is to relax the assumptions classical sampling-based planning methods make about what constitutes a model (e.g., replacing a geometric environment model with sensory experience) and what constitutes a state (e.g., enabling search and planning directly over images). The resulting planner: i) solely operates on information obtainable by the agent on its own through on-board sensing, ii) can adapt and improve its performance with experience.

**Limitations**: The two main assumptions of PALMER about $\pi$ and $\mathcal{D}$ also constitute its two main limitations.

i) Why assume $\mathcal{D}$ (exploration data) is given: Exploration is an exceptionally challenging problem entirely of its own, and a general cure-all approach to exploration without considering the specifics of the problem or the environment may not be efficient or tractable. Therefore, PALMER opts to outsource the collection of $\mathcal{D}$ to an auxiliary method -- **just as all kinematic planning methods (e.g., RRT, PRM) assume a geometric environment model, and all offline RL methods similarly assume an offline exploration dataset.**

ii) Why assume $\pi$ is given:  PALMER is a global planning algorithm, and as such its main focus is **long-term decision making rather than local navigation between close-by states**. We again think that the latter is a challenging problem of its own, especially in those settings where the system dynamics require precise control. This is conceptually the same motivation why robot motion planning literature commonly abstracts away kinematic planning from the downstream control pipeline.

---

### Meta-Review · Area_Chair_oEp4 · 2022-08-26

**Recommendation:** Accept
**Confidence:** Certain

**Metareview:**

The paper presents a method for goal-directed trajectory planning for offline RL on experience replay buffers.  The work builds on classic ideas of sample based planning in robotics (PRM and RRT), but the proposed method is purely experiential and can operate directly with high-dimensional observations instead of a given low-dimensional state space.  The work reduces the presence of false graph connections as compared to earlier techniques.  The method also learns a reachability preserving embedding function that maps observations to a low-dimensional state space.  The experiments show the technique is effective for navigation tasks, and outperforms simpler baselines.

The reviewers found the paper's arguments well motivated (PChv, BFRQ, BnRU), and well written (BFRQ, t3oP), with significant improvements over baselines (PChv, BnRU, t3oP).  The reviewers also had a long list of questions on the limitations of the approach, and how well it connected to the question of planning in a general RL problem.  The author response addressed the reviewers' questions, and described both the strengths and limitations of the proposal.  The reviewers indicated their satisfaction with the response, and their core concerns were addressed.

Four reviewers indicate to accept this paper for its contribution of a new method for a memory-based planning method.  The paper is therefore accepted.


**Award:**

No

---

### Decision · Program_Chairs · 2022-09-14

Accept